# Does the Phytochemical Diversity of Wild Plants Like the *Erythrophleum genus* Correlate with Geographical Origin?

**DOI:** 10.3390/molecules26061668

**Published:** 2021-03-17

**Authors:** Cédric Delporte, Nausicaa Noret, Cécile Vanhaverbeke, Olivier J. Hardy, Jean-François Martin, Marie Tremblay-Franco, David Touboul, Anais Gorel, Marie Faes, Caroline Stévigny, Pierre Van Antwerpen, Florence Souard

**Affiliations:** 1RD3-Pharmacognosy, Bioanalysis and Drug Discovery, Faculté de Pharmacie, Université libre de Bruxelles, Campus Plaine, CP 205/05, 1050 Brussels, Belgium; marie.faes@ulb.be (M.F.); caroline.stevigny@ulb.be (C.S.); pierre.van.antwerpen@ulb.be (P.V.A.); 2Analytical Platform of the Faculty of Pharmacy, Faculté de Pharmacie, Université libre de Bruxelles, Campus Plaine, CP 205/05, 1050 Brussels, Belgium; 3Laboratoire d’Écologie végétale et Biogéochimie, Faculté des Sciences, Université libre de Bruxelles, Campus Plaine, CP 244, 1050 Brussels, Belgium; Nausicaa.Noret@ulb.ac.be; 4Département de Pharmacochimie Moléculaire (DPM), Univiversité Grenoble Alpes, CNRS, 38000 Grenoble, France; cecile.vanhaverbeke@univ-grenoble-alpes.fr; 5Evolutionary Biology and Ecology, Faculté des Sciences, Université libre de Bruxelles, Campus du Solbosch, CP 160/12, 1050 Brussels, Belgium; olivier.hardy@ulb.ac.be; 6Toxalim, Research Centre in Food Toxicology, Toulouse University, INRAE, UMR 1331, PF MetaToul-AXIOM, 31027 Toulouse, France; jean-francois.martin@inrae.fr (J.-F.M.); marie.tremblay-franco@inrae.fr (M.T.-F.); 7CNRS, Institut de Chimie des Substances Naturelles, Université Paris-Saclay, UPR 2301, 91198 Gif-sur-Yvette, France; david.touboul@cnrs.fr; 8Laboratory of Plant Ecology, Department of Plants and Crops, Faculty of Bioscience Engineering, Ghent University, 9052 Ghent, Belgium; AnaisPasiphae.Gorel@UGent.be; 9Department of Pharmacotherapy and Pharmaceutics, Faculté of Pharmacie, Université libre de Bruxelles, 1050 Brussels, Belgium

**Keywords:** eco-metabolomics, plant-omics, plant fingerprint, metabolomics, natural variation, molecular networks, cassane-type diterpenes, specialized metabolites, geographic variation, phytochemical

## Abstract

Secondary metabolites are essential for plant survival and reproduction. Wild undomesticated and tropical plants are expected to harbor highly diverse metabolomes. We investigated the metabolomic diversity of two morphologically similar trees of tropical Africa, *Erythrophleum suaveolens* and *E. ivorense*, known for particular secondary metabolites named the cassaine-type diterpenoids. To assess how the metabolome varies between and within species, we sampled leaves from individuals of different geographic origins but grown from seeds in a common garden in Cameroon. Metabolites were analyzed using reversed phase LC-HRMS(/MS). Data were interpreted by untargeted metabolomics and molecular networks based on MS/MS data. Multivariate analyses enabled us to cluster samples based on species but also on geographic origins. We identified the structures of 28 cassaine-type diterpenoids among which 19 were new, 10 were largely specific to *E. ivorense* and five to *E. suaveolens*. Our results showed that the metabolome allows an unequivocal distinction of morphologically-close species, suggesting the potential of metabolite fingerprinting for these species. Plant geographic origin had a significant influence on relative concentrations of metabolites with variations up to eight (*suaveolens*) and 30 times (*ivorense*) between origins of the same species. This shows that the metabolome is strongly influenced by the geographical origin of plants (i.e., genetic factors).

## 1. Introduction

As sessile organisms, plants produce thousands of secondary metabolites [1,2,3,4] to cope with various environmental constraints [5]. Plant natural products are produced via numerous specialized enzymes that transform central/primary metabolites into secondary metabolites. The latter provide tremendous resources for human therapies [6]. The metabolome is often observed as the “readout” of the physiological status and the bridge between genotype and phenotype [7]. Given the importance of plant secondary metabolism for plant survival and reproduction, and for human health, various studies have been performed to understand the genetic regulation of plant metabolism [8,9]. As metabolites are end products of complex cellular networks of gene expression, protein synthesis and interactions, and are likely under natural selection, they are more accurate measures of the phenotype than transcriptomic or proteomic data alone [10,11,12].

The literature mostly focuses on the description of the metabolome of different species [13,14,15], often crops or model plants, and little is known about variation within wild species. Here, we investigate the variation of genetic basis within two closely-related species, and we compare the levels of intra- and inter-species variation. As our aim is to study natural variation, we focused on wild plants that have not been modified by humans. In this context, *Erythrophleum* genus is an ideal study case of wild undomesticated plants containing original human-bioactive metabolites. *Erythrophleum* (Fabaceae-Caesalpinioideae) is a pantropical woody genus that includes around 15 species with representatives found in North-East Asia, Australia, and Africa [16].

In Africa, *E. suaveolens* (sua) and *E. ivorense* (ivo) are two species found in wild tropical forests. Morphologically they are very similar and both species are often confused, particularly in the absence of reproductive organs, and both are called “tali” by local people [16]. *E. ivorense* is present in high rainfall evergreen and semi-evergreen tropical forests from West Africa to western Central Africa, from Gambia to Gabon. *E. suaveolens* is more widespread, occurring in West, Central and Austral Africa, and tolerates a wider range of climatic conditions. This species is thus found in drier, semi-evergreen forests, in savannahs, and in gallery forests, from Senegal to Sudan and Kenya, as well as south to Mozambique and Zimbabwe [16]. Population genetics research conducted in West and Central Africa showed that each species can be subdivided in three genetic groups occurring in parapatry or allopatry, presumably as a legacy of ancient population fragmentations when the forest cover had contracted due to Quaternary climate changes (glacial episodes) [14].

*Erythrophleum* species have a particular metabolome composed a priori of large amounts of molecules with the specific cassane-type diterpene scaffold [17,18,19]. These are specific scaffolds long recognized for cardiac effects in humans [20]. Cassane-type diterpenes usually named as “cassaine alkaloids” or “*Erythrophleum* alkaloids” were long known as the main specialized metabolites found in the *Erythrophleum* genus [21] whereas the distribution of *nor*-cassane diterpenes is restricted to *Caesalpinia* [22]. Cassaine diterpene has a basic skeleton of the cassane-type (Figure 1). Cassane diterpenoids have been widely studied and their structural diversity is impressive (a SciFinder^n^ search reveals 408 cassane type structures published in 334 references until January 2021 in Chemical Abstracts). The basic cassane type skeleton consists of a tricyclic diterpene with 20 carbon atoms, where the carbon in position C-13 is substituted by an ethyl group and the carbon in C-14 is substituted by a methyl group (Figure 1A). Various biosynthetic pathways have been proposed [23,24,25], in which cassane-type diterpenes are thought to be derived from biosynthetic rearrangements of the pimarane precursor Δ^8^,15,-pimaradiene, itself originating from the cyclization of Δ^8^,15,-labdadienyl pyrophosphate (LDPP) obtained from geranylgeranyl pyrophosphate (GGPP) in the terpenoid biosynthetic pathway (mevalonate pathway) (Figure 1A). The cassane-type backbone would derive from pimarane following migration of the methyl group carried by the carbon in position C-13 to the carbon in position C-14.

In the largest family of cassane-type diterpenes, the cassaine-type ones are a subfamily of molecules which have at the C-13 position a side chain containing a nitrogen atom in various positions. Cassaine was initially isolated from *E. suaveolens* in 1935 [21]. Since then, a large number of different molecules have been described, differing in their substitution patterns at the level of the tricyclic skeleton and of the chain at the C-13 position. Cassaine-type diterpenoids fall into two groups: cassaine-type ester amine diterpenoids (Figure 1B) and cassaine-type amide diterpenoids (Figure 1C) [25,26] with the following particularities:Cassaine-type diterpenoids with an ester amine arm (Figure 1B) result from esterification between the carboxylic group of a cassane-type tricyclic diterpene acid and the alcohol group of an aminoethanol (often *N*-methylethanolamine (CH_3_-NH-CH_2_-CH_2_-OH) or *N,N*-dimethylethanolamine ((CH_3_)_2_N-CH_2_-CH_2_-OH)) [26]. Ethanolamine can be formed by decarboxylation of a serine or by a transamination reaction (exchange of an amine group) between a glycoaldehyde and a glutamic acid [24,25].Cassaine-type diterpenoids with an amide alcohol arm (Figure 1C) follow the same biosynthetic pathway but the last step is likely the binding of a diterpenic acid to *N*-methylethanolamine leading to an alcohol function at the end of the arm [24,25].

In this work, we investigated the molecular diversity of plant metabolism in vivo using leaves from young trees of *E. suaveolens* and *E. ivorense* from different geographic origins but grown in a common garden experiment in Cameroon. Knowing the remarkable secondary metabolites composition of *Erythrophleum*, we investigated whether metabolomics would enable us to discriminate the two *Erythrophleum* species, the intraspecific genetic groups and/or the geographical origins of trees, using liquid chromatography coupled to a high-resolution tandem mass spectrometer (LC-HRMS(/MS)). Moreover, we investigated if particular cassaine-type diterpenoids can be identified as biomarkers of either species and/or origins. For this purpose, LC-HRMS(/MS) data were interpreted by both untargeted metabolomics using the Workflow4metabolomics platform (W4M) [27] and molecular networks using the MZ-Mine and MetGem softwares [28,29]. In fact, Ernst et al. have demonstrated [30] that molecular networks coupled with relevant ecological data provide a robust workflow for investigating plant specialized metabolites in a chemo-evolutionary context. Therefore, we aimed to answer the following questions: (1) do the metabolomes of both *Erythrophleum* species differ in metabolites and/or their relative abundances? (2) do the metabolomes of conspecific trees from different geographic origins differ?

## 2. Results

### 2.1. Metabolomics Analysis

Data analysis using the W4M platform enabled us to detect 1186 features (i.e., one *m*/*z* [mass/charge] value at one retention time) among the 127 samples (composed of three replicates per plant sample). Principal component analysis (PCA) clearly distinguished the two species, *E. ivorense* and *E. suaveolens*, along the first axis (PC1) accounting for 49% of the variance (Figure 2A). This means that despite the morphological similarity of these two species, their phytochemical profiles are clearly distinct. The second axis explained 8% of the variance and was obviously related to intraspecific variation (data not shown), especially in *E. ivorense*. Figure 2A shows that features distributed on the negative values of axis 1 (PC1) are distinctive of *E. ivorense* whereas those on the positive values are specific or more abundant in *E. suaveolens*. Focusing on *E. suaveolens*, PCA plots also show intraspecific variation (axis 2, PC2), especially between northern (CONG, UFA identified by suaN in blue) and southern (LAST, MEK identified by SuaS in green) populations. The high percentage of variance (57%) explained by the first two axes (PC1 + PC2) suggests that hereditary genetic factors predominantly influence the metabolome compared to environmental growth factors as they were cultivated in the same forest concession (common garden). They were wild plants that were not selected, domesticated and crossed [16], so there is a high possibility that the genetics of these species should be very diverse from the adaptation of the wild plant to their geographical origins.

To further study the grouping of samples based on feature similarity, hierarchical clustering was performed without a priori grouping information (Figure 3). A heatmap is a graphical representation of a data matrix where each column represents a sample and each row corresponds to a feature (i.e., one *m*/*z* [mass/charge] value at one retention time) with the ion relative abundance detected in each sample represented by color intensity. In most cases, extraction replicates of a same plant sample were grouped together in the hierarchical clustering dendrogram, and they always belonged to the same sample cluster, which validates our extraction protocol. Rows (features) and columns (samples) of the matrix clustered each in four main groups. Approximately 270 features from the sample cluster 1 are common to all samples. As expected, the two species appeared as two main clusters (sample clusters 1 and 2: *E. ivorense*; and sample clusters 3 and 4: *E. suaveolens).* The two species are distinguished by higher and lower abundances of ions of the feature cluster 2 and 3, respectively, in *E. suaveolens* compared to *E. ivorense*. Unexpectedly, *E. ivorense* samples were split into two subgroups (clusters 1 and 2: *E. ivorense*, named “ivo-Libr 1” and “ivo-Libr 2”) even though they all come from one population. Regarding *E. suaveolens*, the two sample clusters 3 and 4 were not divided according to genetic groups (sua(N) and sua(S)) as would be expected. In contrast, cluster 3 includes the eastern most population named “Ouesso-Sua”, whereas cluster 4 (named “Sua *”) includes all other suaN and suaS samples. These results are coherent with PCA score plot showing the stretching on PC2 axis of PCA of *ivorense* data (Figure 2A) accounted for a group of features located on the negative values of PC2 axis. Obviously, a large set (~18 %) of metabolic features are clearly more abundant in the ivo-Libr 1 subgroup corresponding to sample cluster 1 (red color on the heatmap, feature cluster 4 in sample cluster 1, Figure 3) whereas their concentrations are not variable in the other samples (except one replicate of two samples in *E. suaveolens*).

Considering the clustering organization shown in the heatmap, a PCA analysis including the five populations (Table 1) was performed and plotted (Figure 2B). The 3D-view shows that the four groups (one group cluster two populations) are distributed along the PC1 axis from lower to higher values: light blue for ivo-Libr 2, orange for ivo-Libr 1, Sua * divided in dark blue (=sua (S)) and in green (=sua (N) from Table 1) and finally yellow for Ouesso-Sua. This stretching of populations along PC1 shows that this axis (PC1) includes both inter- and intraspecific variability. The PCA confirms that the sua(N) and sua(S) genetic groups within the Sua * cluster do not differ at the metabolome level.

### 2.2. Molecular Networks

Molecular networks aim to represent the relationships between features based on their structural similarity, which is assessed by similar MS/MS spectra. MS/MS spectra having a strong spectral similarity (cosine score (CS) > 0.55 in our case) are represented as connected nodes. Thus, bonds between nodes appear on molecular networks as soon as the CS is higher than 0.55 and at least ten common fragment ions are detected.

Based on the cassane-type diterpenoid scaffold, there are two possibilities for the bio-synthetic linkage of the arm on C-13 (HO-CH_2_-CH_2_-NH_2_, Figure 1) to form either an amide or an ester (cassane-amide/ester analogs). Either the arm is connected by the alcohol and forms an ester arm (Figure 1B), or the link is made by the amine and in this case an amide is formed (Figure 1C). As a matter of fact, little is known about the cassane diterpenoid MS/MS fragmentation [17,19,24,26,31]. We obtained from the Champy team [18] purified fractions of cassane-type diterpenes from the ester and the amide families. LC-MS/MS analyses were carried out at different collision energies and corresponding MS/MS data were analyzed (see Appendix A for fragmentation description and Figure 4). The main difference between the amide and ester families is the fragmentation of the arm on C-13 of the cassane diterpene cycle. Amide compounds with a terminal alcohol were characterized by a neutral loss of water (−18.0109, δ = 1.15 ppm) and of CH_3_OH (−32.0262, δ = 0.93 ppm). Concerning the ester compounds with a terminal methylamine on the arm, a neutral loss of CH_3_NH_2_ (−31.0422, δ = 12.64 ppm) was observed. It is noteworthy that a common loss of C_3_H_9_NO was also observed for both ester and amide compounds which corresponded to respective ester or amide cut of the arm (−75.068, δ = 5.51 ppm) (Appendix A).

Given the lack of data in MS/MS databases concerning the other cassane-type diterpenes, we have chosen to explore our dataset by using molecular networks. MetGem software [28] was used to visualize networks after a MZmine 2 preprocessing [29]. This work aims at having molecular information on the different clusters, by observing the four subgroups ivo-Libr 1, ivo-Libr 2, Ouesso-Sua and Sua * (Figure 5 and Figure 6 and Table 2). The molecular network was built by analyzing all the MS/MS data and grouping intensities for each feature of each group. The means have been calculated of all the samples within the four subgroups (ivo-Libr1, ivo-Libr2, Ouesso-sua and Sua *). Data analysis revealed different MS/MS sub-clusters and the first and biggest one has been attributed to cassane-type diterpenes (Figure 5 and Figure 6 and Table 2) thanks to Champy’s sample analysis. This cluster illustrating cassane-type diterpenes diversity, is composed of 262 nodes, where 137 and 125 nodes are more present in *E. ivorense* and *E. suaveolens*, respectively. This cassane-type diterpenes cluster was further divided into three sub-clusters named A, B and C illustrated on Figure 5B where principal nodes are labelled by sub-clusters (A, B or C) and a number, example [**A1**] for node 1 in sub-cluster A.

#### 2.2.1. Sub-Cluster A

The first node that we analyzed was more present in *E. suaveolens* and particularly *Sua ** samples. It was the one characterized by *m*/*z* 420.2733 RT = 16.97 min [**A1**] (Figure 6, Table 2) corresponding to the well-known *nor*-cassamine or *nor*-cassamide [17,26,32,33]. However, based on the fragmentation pattern described in Figure 4, nor-cassamine was identified. Indeed, the first fragment, *m*/*z* 389.2370 results from the loss of CH_3_NH_2_.

If the arm was in an amide form, the loss would theoretically be −18.0109 and −32.0262 corresponding to the loss of H_2_O and CH_3_OH, respectively (see Appendix A). Moreover, fragments at *m*/*z* 345.2608 (M-C_3_H_9_NO), corresponding to the *N*-methyl arm loss, *m*/*z* 317.2161 (M-C_3_H_9_NO-CO), *m*/*z* 285.1787 (M-C_3_H_9_NO-CH_3_COOH), *m*/*z* 267.1682 (M-C_3_H_9_NO-CH_3_COOH-H_2_O), and *m*/*z* 257.1853 (M-C_3_H_9_NO-CH_3_COOH-CO) (Figure 4) were in accordance with literature [17].

Based on the exact masses and the well-known fragmentation of the substituted 3-4-cassane-type diterpenes, with the loss of the arm and then the successive losses of the esters in position 4 and then position 3, we were able to propose the other structures. As a matter of fact, a first neighbor to node [**A1**] with a cosine score (CS) of 0.71 was another more representative compound of *Sua *.* The node [**A2**] is characterized by *m*/*z* 406.2569 RT = 16.75 min and corresponds to a new molecule C_23_H_35_NO_5_ (δ = 4.69 ppm). In this case, the first fragment observed was *m*/*z* 345.2062 corresponding to the loss of C_2_H_7_NO arm (without *N*-methyl), and the second *m*/*z* 285.1806, loss of CH_3_COOH as previously described in Figure 4. With this fragmentation pattern, we were not able to define if the arm is linked by an amide or an ester function.

Two other nodes linked to [**A1**] are more abundant in *E. suaveolens* (Figure 5), with *m*/*z* 434.2549 RT = 14.53 min (CS 0.72) and *m*/*z* 434.2855 RT = 17.16 min (CS 0.72) with C_24_H_35_NO_6_ (δ = 2.74 ppm) [**A3**] and C_25_H_39_NO_5_ (δ = 10.59 ppm) [**A4**], respectively. At this collision energy, the first fragments for C_24_H_35_NO_6_ [**A3**] corresponded to *m*/*z* 403.2047 (M-CH_3_NH_2_) and *m*/*z* 359.1844 (M-C_3_H_9_NO), with the loss of the *N*-methyl arm. The fragments *m*/*z* 299.1598 (M-C_3_H_9_NO-CH_3_COOH) and *m*/*z* 281.1675 (M-C_3_H_9_NO-CH_3_COOH-H_2_O) were also observed and attributed to dehydro-*nor*-erythrosuamine as previously described in *E. suaveolens* (synonym: *E. guineense* [32]) and another *Erythrophleum* species, *E. chlorostachys* [32,34]. For C_25_H_39_NO_5_ [**A4**], considering the high CS of 0.74 with [**A1**], we attributed it to cassamine [31,35,36,37,38,39,40,41] that was previously identified in *E. suaveolens* [17,31,32,42].

In this case, during the fragmentation, we have observed fragments at *m*/*z* 371.2058 (M-NH(CH_3_)_2_-H_2_O) and *m*/*z* 345.1959 corresponding to the loss for the dimethyl-amine arm (so an ester form). Fragments *m*/*z* 285.1790 (M-C_4_H_11_NO-CH_3_COOH) and then 257.1769 (M-C_4_H_11_NO-CH_3_COOH-CO) confirmed this possibility.

Interestingly, another node [**A5**] with CS of 0.63 with [**A1**], is more present in *E. ivorense,* and particularly in ivo-Libr 2 with exactly the same fragmentation pattern as [**A3**]. The molecule [**A5**] has *m*/*z* 434.2555 RT = 15.76 min, whereas [**A3**] has *m*/*z* 434.2549 RT = 14.43 min.

#### 2.2.2. Sub-Cluster B

This cluster (Figure 5, Table 2) is linked with the node [**B1**] *m*/*z* 494.2757 RT = 14.44 min, corresponding to C_26_H_39_NO_8_ (δ = 1.74 ppm). This molecule was more present in *Sua ** and is linked to four principal neighbors, three more abundant in *E. suaveolens* ([**B2**], [**B3**] and [**B4**] with CS of 0.72, 0.82 and 0.78, respectively) and one more present in *E. ivorense* [**B5**] with a CS 0.8. The latter is observed four times in the network with similar exact mass, RT and fragmentation patterns and we therefore considered them as only one compound.

For the molecule [**B1**], a fragment *m*/*z* 363.2301 (M-CH_3_NH_2_) was first observed before the loss of the *N*-methyl arm, *m*/*z* 419.2012 (M-C_3_H_9_NO). Consecutive fragments *m*/*z* 401.1922 (M-C_3_H_9_NO-H_2_O), *m*/*z* 359.1797 (M-C_3_H_9_NO-H_2_O-CH_3_COOH), *m*/*z* 299.1569 (M-C_3_H_8_NO-H_2_O-2(CH_3_COOH)), *m*/*z* 281.1476 (M-C_3_H_9_NO-2(H_2_O)-2(CH_3_COOH)) and *m*/*z* 271.1714 (M-C_3_H_9_NO-H_2_O-2(CH_3_COOH)-CO) were also observed. We do not know if CO is in position 6 or 7, but the one in position 6 had already been described in *E. chlorostachys* [43].

In *E. suaveolens,* the molecule [**B2**] *m*/*z* 452.2664 RT = 12.60 min corresponds to C_24_H_37_NO_7_ (δ = 4.7 ppm) and was attributed to a monoacetylated *nor*-cassamine; it was more present in Ouesso-Sua. Indeed, fragments at *m*/*z* 421.2293 (M-CH_3_NH_2_), *m*/*z* 377.1976 (M-C_3_H_9_NO), *m*/*z* 359.1824 (M-C_3_H_9_NO-H_2_O), *m*/*z* 317.1752 (M-C_3_H_9_NO-CH_3_COOH), *m*/*z* 299.1641 (M-C_3_H_9_NO-CH_3_COOH-H_2_O), *m*/*z* 271.1709 (M-C_3_H_9_NO-CH_3_COOH-H_2_O-CO) and *m*/*z* 253.1540 (M-C_3_H_9_NO-CH_3_COOH-2H_2_O-CO) were observed. If both ester and amide arm of cassane-type diterpene have been described [39,44,45,46] however they were never described in *E. suaveolens* leaves.

Molecule [**B3**], 480.2607 RT = 14.33 min, corresponding to C_25_H_37_NO_8_ (δ = 7.32 ppm) was identified. This molecule has not yet been described in the literature but considering the fragmentation pattern and by comparing with neighbors, it likely has a cassane diterpenoid scaffold [**B2**] with the same arm as molecule [**A2**]. Indeed, a first loss of the ester arm was observed at *m*/*z* 463.2232 (−17.0375; M-NH_3_) then m/z 419.2082 (−61.0525, M-C_2_H_7_NO). We then observe the following fragments: *m*/*z* 401.1944 (M-C_2_H_7_NO-H_2_O), *m*/*z* 359.1776 (M-C_2_H_7_NO-CH_3_COOH), *m*/*z* 341.1700 (M-C_2_H_7_NO-CH_3_COOH-H_2_O), *m*/*z* 317.1719 (M-C_2_H_7_NO-CH_3_COOH-H_2_O-CO), *m*/*z* 299.1610 (M-C_2_H_7_NO-2(CH_3_COOH)), and *m*/*z* 281.1509 (M-C_2_H_7_NO-2(CH_3_COOH)-H_2_O), *m*/*z* 271.1687 (M-C_2_H_7_NO-2(CH_3_COOH)-CO), and *m*/*z* 253.1555 (M-C_2_H_7_NO-2(CH_3_COOH)-CO-H_2_O).

The last molecule [**B4**] of this group has *m*/*z* 480.2828 RT = 14.51 min and corresponds probably to C_25_H_37_NO_8_ (δ = 49 ppm). We supposed that this node corresponds to [**B3**] due to the close retention time. This node probably results from the approximate MZmine processing. Indeed, if we consider the theoretical calculation from the *m*/*z* value, the molecular formula is the ester C_22_H_41_NO_10_ (δ = 14.51 ppm). No such cassane is described in literature and we cannot manage to further describe it but we believe that a NH_2_ should occur at the final position as we have the fragment *m*/*z* 419.1992 (M-C_2_H_7_NO).

In *E. ivorense,* molecule [**B5**] *m*/*z* 508.2980 RT = 14.40 min, corresponds to C_27_H_41_NO_8_ (δ = 10.64 ppm). It was a *N*-methyl-cassaine due to the two fragments at *m*/*z* 477.2686 (M-CH_3_NH_2_) and *m*/*z* 433.2176 (M-C_3_H_9_NO). Moreover, fragments at *m*/*z* 373.1963 (M-C_3_H_9_NO-CH_3_COOH), *m*/*z* 313.1796 ((M-C_3_H_9_NO-2(CH_3_COOH)), *m*/*z* 295.1620 (M-C_3_H_9_NO-2(CH_3_COOH)-H_2_O), *m*/*z* 285.1895 (M-C_3_H_9_NO-2(CH_3_COOH)-CO), and *m*/*z* 267.1718 (M-C_3_H_9_NO-2(CH_3_COOH)-CO-H_2_O) confirmed this assignment. As a consequence, molecule [**B5**] has a typical cassane diterpenoid scaffold with an extra methyl in the diterpenoid scaffold compared to [**B1**] but no such molecule has been described in literature. Looking closer to neighbors of [**B5**], we found several molecules with higher molecular masses that were mainly present in *E. ivorense*. 

Molecule [**B6**], *m*/*z* 548.3212 RT = 17.05 min, corresponding to C_30_H_45_NO_8_ (δ = 1.09 ppm), has a similar fragmentation pattern as [**B5**] with a CS of 0.75. Actually, it was identified as the 3β-tigloyl derivative with first fragments corresponding to the loss of the arm, leading to *m*/*z* 517.2759 (M-CH_3_NH_2_) and *m*/*z* 473.2481 (M-C_3_H_9_NO). Then, instead of the double losses of *m*/*z* 60.02 observed for [**B5**] (losses of two acetic acid), we here observed the loss of one acetic acid (*m*/*z* 413.2200; M-C_3_H_9_NO-CH_3_COOH) and the loss of the tigloyl or 3-methylcrotonyl group (*m*/*z* 313.1709, M-C_3_H_9_NO-CH_3_COOH-CH_3_CH=C(CH_3_)COOH or M-C_3_H_9_NO-CH_3_COOH-(CH_3_)_2_C=CHCOOH).

Molecule [**B7**], *m*/*z* 550.3350 RT = 17.28 min, corresponding to C_30_H_47_NO_8_ (δ = 4.45 ppm), was the reduced form of the 3β-tigloyl or 3β-methylcrotonyl derivative. Its fragmentation with the losses of the arm and one acetic acid (*m*/*z* 415.2434; M-C_3_H_9_NO-CH_3_COOH) was followed by the loss of the reduced form of tigloyl (*m*/*z* 313.1745; M-C_3_H_9_NO-CH_3_COOH-CH_3_CH_2_CH(CH_3_)COOH or -(CH_3_)_2_CHCH_2_COOH, −102.0689). As a matter of fact, molecule [**B8**] with *m*/*z* 566.3312 RT = 14.05 min, corresponding to C_30_H_47_NO_9_ (δ= 2.69 ppm), was the hydrate form of molecule [**B6**] with a hydroxyl group instead of the 3β-tigloyl or 3β-methylcrotonyl substitution.

Molecule [**B9**], *m*/*z* 506.3102 RT = 15.65 min, corresponding to C_28_H_43_NO_7_ (δ = 2.04 ppm), was particularly difficult to identify. It is a node neighbor of molecules [**B5**], [**B6**] and [**B8**] (with CS of 0.78, 0.76 and 0.82, respectively), with *N*-methyl arm as suggested by the fragment *m*/*z* 475.2579 (M-CH_3_NH_2_) and *m*/*z* 431.2328 (M-C_3_H_9_NO). However, no fragment was observed around *m*/*z* 371, suggesting the absence of an acetoxy in position 4. However, a fragment at *m*/*z* 313.1701 (M-C_3_H_9_NO-CH_3_CH(OH)CH(CH_3_)COOH or M-(CH_3_)_2_CHCH(OH)COOH, −118.0627) suggested the presence of the hydrate form of the 3β-tigloyl or 3β-methylcrotonyl. We hypothesized that this molecule was 6-oxo-coumingidine previously identified in *E. fordii* [26].

Molecule [**B10**], *m*/*z* 488.2988 RT 15.65 min, corresponding to C_28_H_41_NO_6_ (δ = 3.83 ppm), was also related to molecule [**B8**] with a CS of 0.74 and actually was close to molecule [**B9**] as it possessed the *N*-methyl arm but the fragments *m*/*z* 413.2240 and *m*/*z* 313.1663 (M-C_3_H_9_NO-CH_3_COOH-CH_3_CH=C(CH_3_)COOH or M-C_3_H_9_NO-CH_3_COOH-(CH_3_)_2_C=CHCOOH) suggested the presence of the 3β-tigloyl or 3β-methylcrotonyl. It is noteworthy that [**B11**] is probably an isomer of [**B8**] as we obtained the same fragmentation pattern with a slightly different RT. Interestingly, [**B11**] was mainly present in ivo-Libr 2 and is probably a specific feature of this subgroup.

We also found in this cluster B [**B12**] which is probably a 3-*O*-glucopyrannosyl derivative. Indeed, fragments *m*/*z* 597.2936 (M-CH_3_NH_2_) and *m*/*z* 553.2717 (M-C_3_H_9_NO) were assigned to the *N*-methyl arm. We also observed a fragment *m*/*z* 467.2985 (−161.0338, M-C_6_H_9_O_5_) and a second-high fragment *m*/*z* 331.1976 (M-C_3_H_9_NO-CH_3_COOH-C_6_H_10_O_5_) corresponding to the loss of both the 4-acetoxy- and the 3-*O*-glucopyrannosyl groups. The fragments *m*/*z* 313.1860 ((M-C_3_H_9_NO-CH_3_COOH-C_6_H_10_O_5_-H_2_O) and *m*/*z* 285.1898 ((M-C_3_H_9_NO-CH_3_COOH-C_6_H_10_O_5_-H_2_O-CO) were also observed.

Among other compounds present in *E. ivorense*, [**B13**] with *m*/*z* 392.2802 RT = 13.27 min, corresponding to C_23_H_37_NO_4_ (δ = 1.70 ppm), is a typical precursor of the cassane-type diterpene with the *N*-methyl arm, *nor*-cassaine, as suggested by the fragment *m*/*z* 361.2372 (M-CH_3_NH_2_) and *m*/*z* 317.2101 (M-C_3_H_9_NO) and loss of water molecules (*m*/*z* 299.2101; M-C_3_H_9_NO-H_2_O / *m*/*z* 281.1906; M-C_3_H_9_NO-2H_2_O) or a carbonyl (*m*/*z* 289.2144; M-C_3_H_9_NO-CO). The exact same pattern was observed for [**B14**], *m*/*z* 392.2789 RT = 12.66 min with the loss of the *N*-methyl arm (*m*/*z* 317.2199 (M-C_3_H_9_NO) and loss of one water molecule (*m*/*z* 299.2222; M-C_3_H_9_NO-H_2_O), a carbonyl (*m*/*z* 289.2249; M- C_3_H_9_NO-CO) or both (*m*/*z* 271.2057; M-C_3_H_9_NO-CO-H_2_O).

Molecule [**B15**] is also abundant in *E. ivorense* and was previously identified as coumingidine in *E. fordii* [26], with *m*/*z* 492.3314 C_28_H_45_NO_6_ (δ = 1.15 ppm) RT = 15.83 min. Indeed, the fragments *m*/*z* 461.2905 (M-CH_3_NH_2_) and *m*/*z* 417.2622 (M-C_3_H_9_NO) followed by *m*/*z* 389.2689 (M-C_3_H_9_NO-CO), *m*/*z* 299.2037 (M-C_3_H_9_NO-CH_3_CH(OH)CH(CH_3_)COOH or -(CH_3_)_2_CHCH(OH)COOH), *m*/*z* 281.1868 (M-C_3_H_9_NO-CH_3_CH(OH)CH(CH_3_)COOH or -(CH_3_)_2_CHCH(OH)COOH-H_2_O), *m*/*z* 253.1981 (M-C_3_H_9_NO-CH_3_CH(OH)CH(CH_3_)COOH or -(CH_3_)_2_CHCH(OH)COOH-H_2_O-CO) were observed. Surprisingly, [**B16**] was identified as a C_28_H_45_NO_6_
*m*/*z* 478.3067 (δ = 20.1 ppm) RT = 15.61 min. Mainly present in *E. suaveolens* samples, it is related to [**B15**] and only differs by the presence of a primary amine on the arm. Indeed, the first loss was the amine arm (*m*/*z* 417.2685; M-C_2_H_7_NO) followed by the loss of the 3β-hydroxytigloyl or 3β-hydroxyméthylcrotonyl (*m*/*z* 299.2079; M-C_2_H_7_NO-CH_3_CH(OH)CH(CH_3_)COOH or -(CH_3_)_2_CHCH(OH)COOH-H_2_O).

Molecule [**B17**], *m*/*z* 378.2625 RT = 13.10 min, corresponding to C_22_H_35_NO_4_ (δ = 3.67 ppm) was identified as a new compound. Beside the neutral loss of NH_3_ (*m*/*z* 361.2242, M-NH_3_) and loss of the amine arm (*m*/*z* 317.2155; M-C_2_H_7_NO), fragments at *m*/*z* 299.2056 (M-C_2_H_7_NO-H_2_O), *m*/*z* 281.2047 (M-C_2_H_7_NO-2H_2_O) and *m*/*z* 271.2099 (M- M-C_2_H_7_NO-H_2_O-CO) were also observed. We hypothesized that [**B17**] is a demethylated cassaine which was mainly present in *E. suaveolens*.

#### 2.2.3. Sub-Cluster C

Finally, a subgroup of compounds was mainly expressed in Ouesso-Sua. The compound [**C2**] with a *m*/*z* 452.4476 RT = 11.17 min had a low abundance and the structure was not determined. However, several compounds in this zone are cassane diterpenes, for example, [**C1**], *m*/*z* 452.2636 RT = 11.17 min is C_24_H_37_NO_7_ (δ = 1.50 ppm). Fragments of [**C1**] *m*/*z* 421.2360 (M-CH_3_NH_2_) and *m*/*z* 377.1924 (M-C_3_H_9_NO) were attributed to the loss of the *N*-methyl arm with *m*/*z* 317.1760 (M-C_3_H_9_NO-CH_3_COOH), *m*/*z* 299.1617 (M-C_3_H_9_NO-CH_3_COOH-H_2_O), *m*/*z* 281.1528 (M-C_3_H_9_NO-CH_3_COOH-2H_2_O), *m*/*z* 271.1644 (M-C_3_H_9_NO-CH_3_COOH-H_2_O-CO), *m*/*z* 253.1600 (M-C_3_H_9_NO-CH_3_COOH-2H_2_O-CO). This compound was an isomer of [**B2**]. A neighbor of [**C1**] is [**C2**], *m*/*z* 438.2501 RT = 10.97 min is C_23_H_35_NO_7_ (δ = 3.36 ppm), was a demethylated form of [**C1**]. Indeed, *m*/*z* 377.1923 (M-C_2_H_7_NO) were attributed to the loss of the amine arm while *m*/*z* 317.1731 (M-C_2_H_7_NO-CH_3_COOH), *m*/*z* 299.1594 (M-C_2_H_7_NO-CH_3_COOH-H_2_O), *m*/*z* 281.1534 (M-C_2_H_7_NO-CH_3_COOH-2H_2_O), *m*/*z* 271.1695 (M-C_2_H_7_NO-CH_3_COOH-H_2_O-CO). A final compound that illustrate the sub-cluster C is [**C3**], *m*/*z* 466.2787 RT = 11.31 min is C_25_H_39_NO_7_ (δ = 2.64 ppm), was a *N,N*-dimethylamine form of [**C1**]. Indeed, fragments *m*/*z* 421.2158 (M-(CH_3_)_2_NH) and *m*/*z* 377.2053 (M-C_4_H_11_NO) were attributed to the loss of the *N,N*-dimethyl arm. Then, fragments could also be attributed including *m*/*z* 317.1684 (M-C_4_H_11_NO-CH_3_COOH), *m*/*z* 299.1638 (M-C_4_H_11_NO-CH_3_COOH-H_2_O), *m*/*z* 281.1534 (M-C_2_H_7_NO-CH_3_COOH-2H_2_O), *m*/*z* 271.1695 (M-C_2_H_7_NO-CH_3_COOH-H_2_O-CO).

#### 2.2.4. Other Networks

The molecular network analysis had also emphasized the diversity of polyphenol derivatives (Figure 6). Based on GNPS analogs search [47,48,49,50], catechin derivatives for 2nd cluster and flavonoid-glycosides for 4th cluster were mainly observed in Sua*. In contrast, quercetin derivatives were observed in ivo-Libr 1 for 4th cluster. Finally, a trihydroxyflavanone derivative was particularly present in *E. suaveolens* for 8th cluster (framed in Figure 6).

### 2.3. Species-Specific Metabolite Features and Identification

Biosigner analysis was performed on W4M to determine which features have significantly different abundances in the two species (ivo and sua), in order to identify species-specific metabolites. Two ions were highlighted according to a minimum of two of the three distinct classifier algorithms (PLS-DA, RF, and SVM methods). This metabolomics variable selection evaluated the relevance of the variables for the prediction performances of the classifier [51]. Three features highlighted in Figure 4 were kept: [**BS1**] had a *m*/*z* 450.2492 and RT = 13.21 min and was a marker of *E. suaveolens*. Based on the molecular network, the putative cassane-type diterpenes molecular formula was C_24_H_35_NO_7_ (δ = 2.07 ppm) and we assigned this molecule to 3β-hydroxy-4-acetoxy-6-oxo-*nor*-cassamine. Actually, we were able to observe the following fragments *m*/*z* 419.2053 (M-CH_3_NH_2_), *m*/*z* 375.1831 (M-C_3_H_9_NO), *m*/*z* 357.1632 (M-C_3_H_9_NO-H_2_O), *m*/*z* 315.1587 (M-C_3_H_9_NO-CH_3_COOH), *m*/*z* 297.1476 (M-C_3_H_9_NO-CH_3_COOH-H_2_O), and *m*/*z* 269.1518 (M-C_3_H_9_NO-CH_3_COOH-H_2_O-CO). The second one, [**BS2**] had a *m*/*z* 490.3158 and RT = 19.28 min and was a marker of *E. ivorense* with a putative formula C_28_H_43_NO_6_ (δ = 1.05 ppm). As described before, we have noticed that several molecules in *E. ivorense* (called “heavy cassanes”) have higher molecular weights than the ones in *E. suaveolens*. Fragments *m*/*z* 459.2685 (M-CH_3_NH_2_), *m*/*z* 415.2507 (M-C_3_H_9_NO), *m*/*z* 315.1960 (M-C_3_H_9_NO-CH_3_CH=C(CH_3_)COOH or M-C_3_H_9_NO-(CH_3_)_2_C=CHCOOH), *m*/*z* 297.1834 (M-C_3_H_9_NO-CH_3_CH=C(CH_3_)COOH-H_2_O or M-C_3_H_9_NO-(CH_3_)_2_C=CHCOOH-H_2_O), *m*/*z* 279.1725 (M-C_3_H_9_NO-CH_3_CH=C(CH_3_)COOH-2H_2_O or M-C_3_H_9_NO-(CH_3_)_2_C=CHCOOH-2H_2_O), and *m*/*z* 269.1873 (M-C_3_H_9_NO-CH_3_CH=C(CH_3_)COOH-H_2_O-CO or M-C_3_H_9_NO-(CH_3_)_2_C=CHCOOH-H_2_O-CO) allowed us to assign this molecule to the structure 3β-tigloyl or 3β-methylcrotonyl-6-hydroxy-*nor*-cassamine, a heavy cassane diterpene specifically found in *E. ivorense*. The third one, [**BS3**] had a *m*/*z* 506.3113 and RT = 16.31 min with a putative structure of C_28_H_43_NO_7_ (δ = 0.14 ppm). It is the same compound as [**B9**] and was accordingly assigned as a 3β-hydroxytigloyl or 3β-hydroxymethylcrotonyl-6-oxo-*nor*-cassamine, another heavy cassane diterpenoid found in *E. ivorense*.

## 3. Discussion

The use of plants by human populations has an important chemical aspect, which can mediate complex people–plant relationships [42]. The perennial plants of *Erythrophleum* genus are being threatened by over-exploitation due to their critical roles in the medicinal applications (in particular for *E. fordii and E. lasianthum*), especially for their high content of cardiac alkaloids [20].

We studied this wild plant, which man has not influenced a priori by cultivation or other selections and used samples from different origins but grown in a common garden experiment to minimize metabolomics differences due to environmental effects. We have assembled a homogeneous pool of 44 mature leaf samples from two species of *Erythrophleum* (*E. ivorense* 23 samples, *E. suaveolens* 21 samples). This wild plant naturally grows in tropical Africa and the two focal species are difficult to identify in the field due to their morphological similarity.

Our untargeted metabolomics investigation obviously discriminates the species (Figure 2A) but we were also able to discriminate intra-specific variations, i.e., geographical origins (Figure 2B). Species discrimination was expected and the clustering of populations of *E. ivorense* and *E. suaveolens* is obvious even if botanical identification is tricky. On the other hand, it is interesting to note that genetically-based variations were detectable on extracts from leaves from trees after planting seeds of various geographical origins. Effectively, seeds were from different genetic and geographical origins but all were cultivated in a common garden experiment (Figure 1). A maternal effect, i.e., the fact that individuals could be influenced by their mother’s local environment, cannot account for the grouping of geographical origins as we used several mother plants for each geographic origin. Furthermore, we did not observe any block effect within the plantation on the metabolome (results not shown, see Table 1 for block allocation), illustrating the limited effect of the growing environment. This observation is particularly true when genetic parameters are consistent; i.e., Ouesso-Sua samples were well gathered according to their genetic filiation. In fact, the genetic microsatellite markers allowed us to identify the species (always) and often to confirm or to identify the family of a specimen or population of origin, especially in *E. suaveolens* where our markers are more polymorphic. As a consequence, heatmap (Figure 3) and PCA (Figure 2B) confirm it from a metabolomics point of view.

In *Erythrophleum*, reproduction occurs mainly by cross-fertilization so that genetic differentiation is not very strong between families and each family may contain a good genetic diversity. In the case of *E. ivorense*, collecting areas were geographically closed (Libreville area) and the non-specific gene breeding is more possible. As a consequence, neutral genetic markers are enabled to discriminate families. Heatmap (Figure 3) and PCA (Figure 2B) clearly separated two sub-clusters ivo-Libr 1 et ivo-Libr 2 in *E. ivorense*, meaning that the metabolome is influenced by either genetic considerations not measured by genetic markers or by environmental factors. However, although the leaflets of all the studied trees were sampled the same day, seeking only mature leaflets, we cannot exclude that part of their metabolomic variation was influenced by differences in leaf maturity or by uncontrolled stress factors.

Within *E. suaveolens*, the two metabolomic clusters identified separated samples from the easternmost population from the other ones. This correlation cannot result from direct environmental growth effects on the metabolome because trees were grown in a common garden experiment, demonstrating a genetic determinism. Interestingly, the expression of this genetic variation through metabolites did not coincide with the main genetic macrogeographic variation detected using neutral microsatellite markers which highlights a north-south differentiation [16]). If metabolomic variation is non-neutral, which is likely for molecules that are costly to produce, it is possible that natural selection contributed to generate a geographic pattern in response to local adaptation pressures, not visible using neutral markers. Within *E. ivorense*, the two metabolomic clusters are not associated with different geographic origins. Here, we cannot assess whether this metabolomic variation results from a genetic determinism showing local polymorphism or from uncontrolled environmental factors. We would have needed several samples per tree, possibly taken at different periods, to be able to tease apart environmental and genetic determinism.

As matter of fact, metabolomics allowed us to classify the samples according to their species and, to some extent, according to geographical origin. In this context, it was interesting to identify the discriminant metabolites.

Heatmap feature clustering is convenient for the first step of annotation as it allows identifying the features—metabolites—of interest. For example, in subcluster ivo-Libr 1, features cluster 4 (Figure 3) is over-expressed. Then molecular network construction is of high interest to identify and to annotate those metabolites as this method allows linking MS features and molecular chemistry (Figure 5 and Figure 6). Cassane-type diterpenes, which are original chemical signatures of *Erythrophleum* genus [24] are well adapted to annotation by molecular networks. Indeed, with their diterpenoids tri-cyclic ring (*m*/*z* 255) substituted by ester, amide, acetoxy or hydroxyl, they have common fragments and neutral losses that allowed in the present study to MetGem to create a well-defined cluster (Figure 5). Moreover, and as several cassane-type diterpenes are described in literature, attempts of molecular assignments are possible based on the well-known structures. First, all the cassane-type diterpenes found were derived from cassamine and as a consequence, carried an amine esterified arm with an ester linkage. No amide derivatives with an alcohol have been identified and various substitutions have been observed on the diterpene cycle. Moreover, and thanks to the molecular network, we found 19 new cassane diterpenes (Table 2). Second, we noticed that *E. ivorense* was characterized by complex 3β-alkyl compound with tigloyl, methylcrotonyl or o-glucosyl derivatives. They are also abundant in the extract when comparing the signal abundance by mass spectrometry (Figure 6, bottom). This statement was also confirmed by Biosigner analysis comparing *E. ivorense* and *E. suaveolens* populations that showed those compounds were significantly higher in *E. ivorense* and consequently are a chemical signature of this species.

Regarding compounds that discriminate ivo-Libr 1 from ivo-Libr 2, only one compound [**B11**] in the cassane-type diterpene cluster can be emphasized. However, in cluster 4, quercetin analogs seemed present in higher amount in ivo-Libr 1. The direct consequence, is the fact that *E. suaveolens* is characterized by cassane-type diterpene with lower masses. For instance, 3β-acetate-4-acetoxy derivatives are well represented in Sua * subgroup while Ouesso-Sua is characterized by 3β-hydroxy-4-acetoxy derivatives. Moreover, the other clusters in the molecular network showed that Sua * can be differentiated by the presence of catechin derivatives (cluster 2), mono- or di-*O*-glycoside-flavonoids (cluster 4) and a trihydroxyflavanone derivative (cluster 8) (Figure 6, top).

To sum up, this study has been carried out on wild plants where the seeds are collected in a wide area and grew in a homogenous environment. Metabolomics has demonstrated its strength to properly discriminate the species and some sub-groups. It is complementary to genetic analysis using microsatellite markers to discriminate groups. Moreover, metabolomics conserves the markers related to genetics and the geographical origins despite the fact that the growing was a new controlled environment (Pallisco) on wild plants.

We were unable to conclude whether the genetic is solely the determinant or if the geographical origin could also have an impact—by epigenetic perhaps. It is also difficult to link the different metabolites with the genetic diversity as the chemical pathways are not totally elucidated for cassane-type diterpenes. Nevertheless, this study demonstrates that metabolomics could properly separate genetic variants (species, populations) of plant seeds even if they grew in the same environment which could help biologists to properly characterize plants from unknown origin. It remains to be demonstrated that these data are stable in different environments and over time. This study offers a better comprehension of the chemical diversity and content in key secondary metabolites with pharmaceutical interest. This study could be useful to help in the seed origin selection for reasonable culture of this threatened species.

## 4. Materials and Methods

### 4.1. Field Sampling

In 2007, seeds of *E. suaveolens* and *E. ivorense* were collected under 30 trees (‘families’) in five sites of Cameroon, Gabon and the Republic of Congo, and were conserved dried. In March 2008, the seeds were soaked 10–15 min in a 90% sulfuric acid solution to break their dormancy. They were then planted separately in two liters polyethylene bags filled with potting soil in the nursery of the logging company Pallisco, in southeastern Cameroon (3°28′ N, 13°34′ E). In March 2009, most seedlings were 30–70 cm high and 20 seedlings from each of 20 families were transplanted with a spacing of 3 m on a cleared horizontal parcel of c. 0.4 ha near the Pallisco headquarter to monitor their growth in identical environmental conditions (http://www.dynaffor.org/ (accessed on January 2021)). This common garden experiment was made of 20 rectangular blocks (12 × 15 m) and each family was represented once per block (random block design). The site is located on Ferralsols at 650 m asl and the climate is Guinean equatorial (pluviosity: 1550–1750 mm/yr; mean monthly temperature: 22–24 °C). The two first years, when a seedling died it was replaced by another seedling from the nursery. In January 2015, trees had reached a mean height of c. 11 m and c. 10% had died since the transplantation. Ten mature leaflets (leaves) were then sampled in four blocks named A to D on 23 trees belonging to *E. ivorense* and 21 trees belonging to *E. suaveolens* (Table 1). Leaflets of each individual were air-dried after harvest for at least 72h and were not exposed to sun. Trees belonging to *E. suaveolens* came from three geographical origins (populations) and two genetic groups—called Sua(S) and Sua(N) (as determined by neutral microsatellite markers) [16] (Table 1). The purpose of growing all plants together in the same plot (common garden experiment) was to assure that all will be exposed to the same environmental conditions (same soil factors, sun exposure, wind, etc.). Consequently, the metabolomic differences we might observe would mainly result from genetically-based differences and not from environmental effects.

The experimental sampling design and the metabolomics carried out in the current study are illustrated in Figure 2. Data were first preprocessed and statistically analyzed with the Workflow4Metabolomics (W4M) platform [27,53]. MS² data were preprocessed in MZmine 2 and molecular networks were constructed in MetGem software [28]. The complete workflow allowed us to perform a holistic identification of samples.

### 4.2. Controlling the Identification and Origin of Samples

As leaves of both species are morphologically indistinguishable, samples were genotyped with nine microsatellites to check species identity following Duminil et al. [16] (data not shown). These data also allowed checking the coherence of geographical origin. It appeared that seeds sampled below a same tree (called ‘family’ here) are not always sibs (results not shown) because secondary seed dispersal is frequent in *Erythrophleum* [54]. Therefore, the smallest unit to group samples and check for potential genetic effects was the population of origin.

### 4.3. Metabolomics Analyses

LC-MS quality formic acid (FA), acetonitrile (ACN) and methanol were purchased from Sigma Aldrich (Steinheim, Germany). High purity water was prepared using a Milli-Q system from Millipore (Bedford, MA, USA).

#### 4.3.1. Sample Preparation

Each dry sample (leaves) was grounded before being extracted on batches of 15 mg of powdered leaves in 1.5 mL of pure methanol for 5 min in a 55 kHz ultrasonic bath at room temperature. Three extraction replicates per sample were performed. Samples were filtered on 0.22 µm membrane and stored in −20 °C before analysis.

#### 4.3.2. LC-HRMS(/MS) Analysis

Analyses were performed using a 1200 series rapid resolution liquid chromatograph (RRLC) coupled to a 6520 series electrospray ionization (ESI)-quadrupole time-of-flight (QTOF) high-resolution mass spectrometer (HRMS) from Agilent Technologies (Waldbronn, Germany). Compound separation was performed using a Poroshell 120 EC-C18 column (2.1 × 100 mm, 2.7 μm particle size, Agilent Technologies, Palo Alto, CA, USA). The column temperature was set at 40 °C. The mobile phases used in all experiments were composed of water acidified with 0.1% formic acid (FA) (solvent A) and acidified acetonitrile (0.1% FA) (solvent B). The applied gradient was as follows: 0 min, 5% B; 0–20 min, 75% B; 20–25 min, 75% B; in 1 min back to 5% B; and 26–30 min, 5% B; post-run 5 min at 0.2 mL/min. ESI-QTOF parameters were as follows for simple MS analysis: positive mode, 2 GHz resolution, MS scan range 100–1600 *m*/*z* at 2 spectra/s, drying gas temperature 350 °C, drying gas flow 9 L/min, nebulizer pressure 50 psi, capillary voltage 4000 V, Fragmentor 175 V. Nitrogen was used as nebulizer gas. Continuous infusion of two reference ions, respectively *m*/*z* 121.050873 and 922.009798, was performed for QTOF continuous calibration. Data acquisition and analysis were carried out by MassHunter Acquisition^®^ software for QTOF (Version B.04 SP3) and MassHunter Qualitative Analysis^®^ (Version B.07) software (both from Agilent Technologies). Samples were randomly analyzed in one batch. The same quality control (QC) sample (=mix of all samples) was injected throughout the run after every ten samples approximately for control and blanks (water with 0.1% FA) were also injected throughout the run.

For molecular network and standard analysis, a LC-autoMS/MS analysis was performed with the following conditions for the ESI-QTOF: MS scan range 100–1600 *m*/*z* at 4 spectra/s; MS inclusion list: 250–900 *m*/*z*; MS/MS scan range 100–1600 at 3 spectra/s; isolation width: medium mode (≈4 *m*/*z*); fixed collision energy was 25 eV for the molecular network and 5, 25, 50 eV for the standard analysis; max precursors at 3/cycle; threshold 2000 absolute intensities.; Precursor abundance based scan speed at 25,000 counts/spectrum with MS/MS accumulation time limit; active exclusion after 3 spectra released after 0.5 min; precursor sort by abundance for charges of 1 or 2.

#### 4.3.3. Metabolomics Data

Agilent format “.d” data were converted to “.mzXML” format using the ProteoWizard MSConvert tools (Version 3.03.9393, 64-bit) with following parameters: binary encoding precision = 32 bit, write index: yes, use zlib compression: yes, TPP compatibility: yes, with the Peak Picking filter option with only MS level 1.

#### 4.3.4. Data Processing and Chemometrics on W4M

Preprocessing (filtration, peak identification, peak grouping and smoothing, retention time correction, integration, annotation), normalization, quality control (metabolites correlation analysis and determination of batch correction), statistical analysis (univariate testing and multivariate modeling) were conducted using the Galaxy workflow4metabolomics W4M (http://workflow4metabolomics.org) [27]. The entire analysis has been processed on 170 samples (samples including extraction replicates + blank + Quality Control samples).

#### 4.3.5. Statistical Analyses

Heatmaps, principal component analysis (PCA) and partial least squares-discriminant analysis (PLS-DA) were performed to visualize the distribution of metabolite variability using W4M platform. The Biosigner tool was used to select the variables which are significant and best distinguish two groups of samples [51] on the W4M platform. Three binary classifiers have been run in parallel, namely partial least squares-discriminant analysis (PLS-DA), random forest and support vector machines (SVM).

#### 4.3.6. Data Preprocessing with MZmine 2 and Molecular Network Analysis with MetGem

mzXML files of autoMSMS analyses were further processed using MZmine 2.53 software [29]. Range time was filtered on 5–24 min as it is the part of interest in the chromatograms. Mass detection was performed by fixing the noise level at 2000 for MS1 and 50 for MS2. ADAP Chromatogram Builder was used to build chromatograms with a minimum group size of three scans, a group intensity at 50 and a minimum intensity at 50. The *m*/*z* tolerance was fixed at 0.008 or 24 ppm. Wavelet (ADAP) method was used for deconvolution with a S/N at 8, a minimum feature height at 500, a coefficient/area threshold at 10, a peak duration range of 0.01–0.50 and a retention time (RT) wavelet range of 0.01–0.09 min. This deconvolution was performed with a *m*/*z* range for MS2 scan pairing of 0.02 Da and a RT range for MS2 scan pairing of 0.25 min. Isotopic peak grouper was then performed using the following parameters: *m*/*z* tolerance = 0.008 amu or 24.0 ppm, RT tolerance = 0.25 min, maximum charge = 1. Peak list was then generated using join aligner for data alignment with the following parameters: *m*/*z* tolerance = 0.08 amu or 24 ppm, weight for *m*/*z* = 75, RT tolerance = 0.5 min, weight for RT = 25. Gal filler option was performed using the same RT and *m*/*z* range method with a *m*/*z* tolerance at 0.008 amu or 24.0 ppm. A filter to keep only peaks with MS2 scan (GNPS), minimum peaks in a row of 1 and minimum peaks in an isotope pattern of 1. Data were then exported as .mgf files for spectra and .csv files for metadata information (intensities, RT …). We used .csv file to calculate means of areas of MS1 for each feature of each sample in the same group observed in heatmap (Figure 4). MetGem 1.3.4 software was used for building the molecular networks (https://metgem.github.io/, [28]). Data “.mgf” and “.csv” files were imported using standard parameters (*m*/*z* tolerance = 0.02; minimum matched peaks = 4. Networks were then created where edges were filtered to have a cosine score (CS) above 0.55, maximum neighbor number (top K) of 10 and a max. connected component size of 1000.

## Figures and Tables

**Figure 1 molecules-26-01668-f001:**
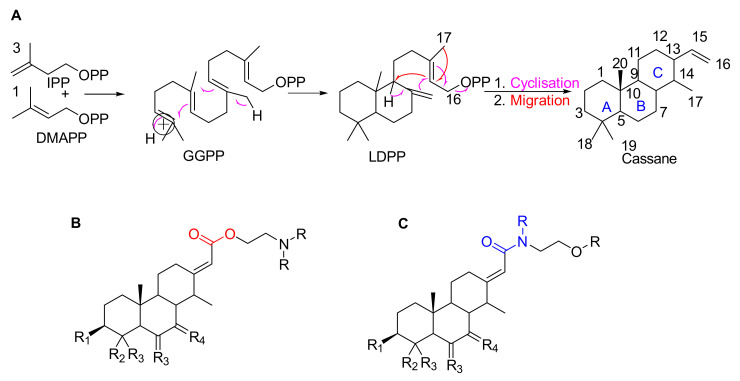
(**A**) Possible biosynthesis pathways for basic skeletons of cassane-types diterpenoids. Three molecules of isopentenyl pyrophosphate (IPP) react with a dimethylallyl pyrophosphate (DMAPP) which form geranylgeranyl pyrophosphate (GGPP) in the mevalonate pathway. The latter rearrange into Δ^8^,15-labdadienyl pyrophosphate (LDPP) which itself rearranges into cassane-type diterpene via cyclisation and migration. (**B**,**C**) Basic skeletons of cassane-type diterpenes with (**B**) an ester or (**C**) amide arm where R substituents are H, Me or any chain and RX.

**Figure 2 molecules-26-01668-f002:**
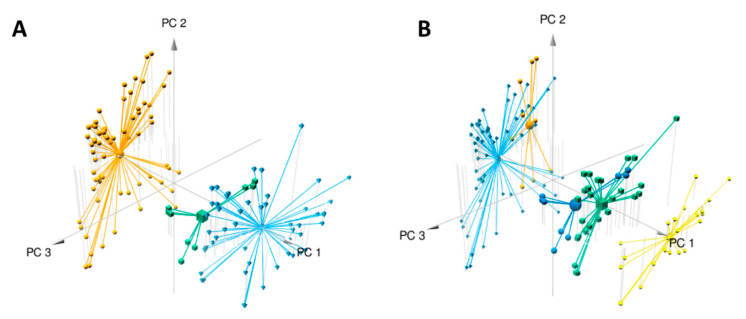
Multivariate modeling using PCA of metabolomics variation in samples from *Erythrophleum* species (*E. ivorense* and *E. suaveolens*). The three first components are plotted and percentages of variance explained by PC1, PC2 and PC3 are respectively 49%, 8% and 7%. R^2^X of the PCA model is 0.648. (**A**) Samples are colored corresponding to the three groups based on species *E. ivorense* (ivo) in orange, *E. suaveolens* North (sua(N)) in blue and *E. suaveolens South* (sua(S)) in green (see Table 1 for details). (**B**) The PCA was drawn with groups identified in the heatmap as a grouping factor (see Figure 3). In light blue ivo-Libr 2, in orange ivo-Libr 1, in green + dark blue Sua * (sua(N) + sua(S) see Table 1) and in yellow Ouesso-Sua.

**Figure 3 molecules-26-01668-f003:**
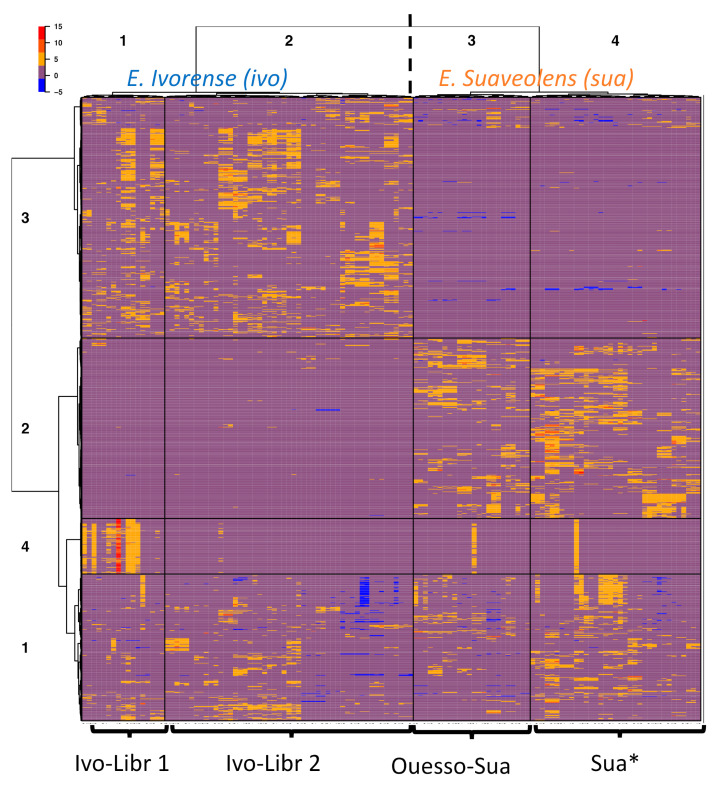
Heatmap clustering representation of metabolic diversity in two *Erythrophleum* species. Samples are in columns and features in rows. Hierarchical clustering splits *E. ivorense* (ivo) into 2 clusters named ivo-Libr 1 (=cluster 1) & ivo-Libr 2 (=cluster 2), and *E. suaveolens* (sua) is splitted into 2 clusters named Ouesso-Sua (=cluster 3) and Sua * (=cluster 4).

**Figure 4 molecules-26-01668-f004:**
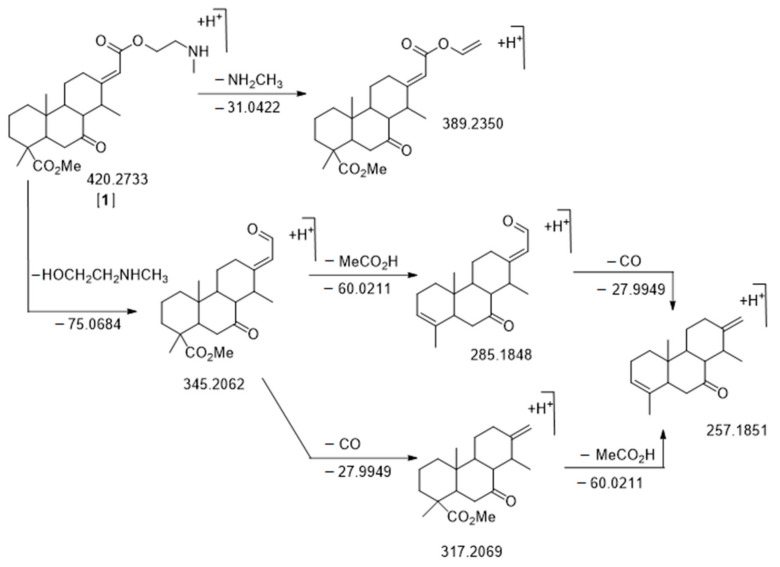
Fragmentation pattern of C_24_H_37_NO_5_
*m*/*z* 420.2733 RT = 16.97 min, *nor*-cassamine [**A1**]. The *m*/*z* values of neutral loss are calculated while the *m*/*z* values of fragments are observed. All δ ppm are under 20 ppm.

**Figure 5 molecules-26-01668-f005:**
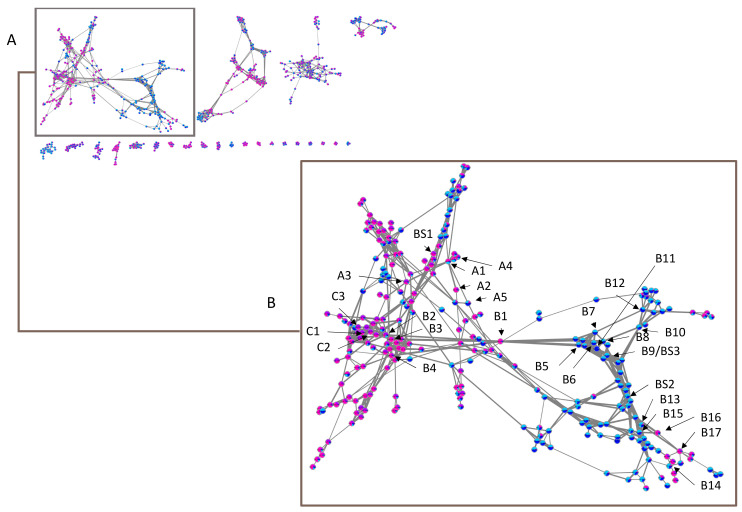
(**A**) Entire molecular network of *Erythrophleum* samples and (**B**) a zoom on cassane-type diterpene network with indications of annotated nodes. Alphanumeric values A, B and C correspond to subclusters mentioned in Table 2 and BS to annotated features assigned by Biosigner algorithm. Pink in nodes represents proportion of Sua *, violet of Ouesso-Sua, light blue of ivo-Libr1 and dark blue of ivo-Libr2 subgroups.

**Figure 6 molecules-26-01668-f006:**
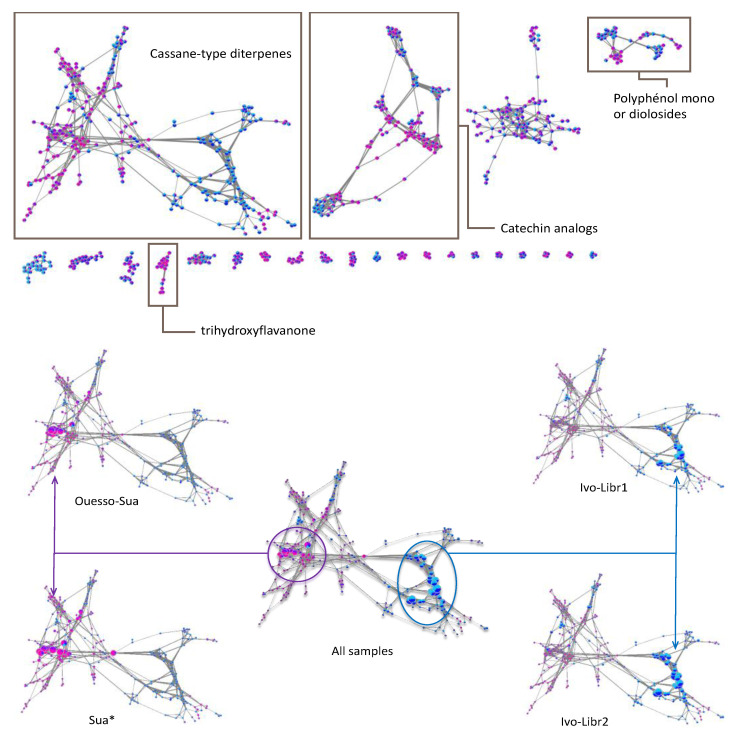
On the (**top**), attribution of the different sub-clusters using GNPS databases, on the (**bottom**), focusing on the cassane-type diterpenes cluster where nodes sizes are proportional of the abundances. Each node is a pie-chart with Sua * in pink, Ouesso-Sua in violet, ivo-Libr1 in light blue, ivo-Libr2 in dark blue.

**Figure 7 molecules-26-01668-f007:**
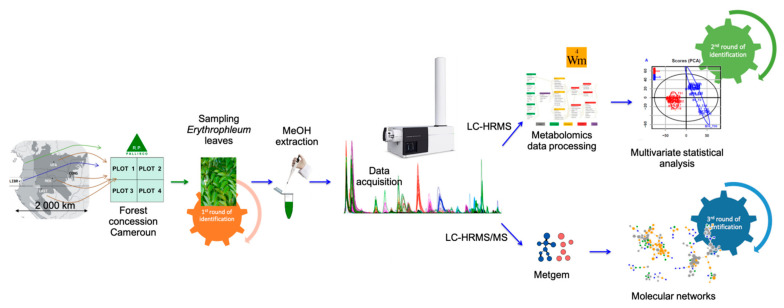
Experimental sampling design and overview of the workflow followed for metabolomics analysis of *Erythrophleum* leaves. *E. suaveolens* and *E. ivorense* from five populations were grown in 4 plots in a forest concession in Cameroon. Leaves of 44 trees were sampled, air-dried and metabolites were extracted with methanol (MeOH). Liquid chromatography coupled to electrospray ionization high-resolution mass spectrometry (LC–ESI-HRMS(/MS)) was performed to characterize the metabolome diversity. Data analysis was completed by exploration using molecular networking.

**Table 1 molecules-26-01668-t001:** Geographic origins and genetic groups of the *Erythrophleum* plant material analyzed for metabolomics diversity in the present study. Location of plants in the four experimental blocks and the number of individuals per origin (*n*) are detailed in the last two columns.

Species	Population/ Country (*)	GPS Coordinates (Long/Lat)	Genetic Groups (**)	Block Abbreviation (Number of Individuals)	*n*
*E. ivorense*	Libreville/C	9.3727/0.6187	Ivo	A(5), B(6), C(7), D(5)	23
*E. suaveolens*	Ouesso/RC	16.4892/1.2926	Ouesso-Sua (N)	A(4), 2B(1), C(2), D(3)	10
	UFA30/C	13.9128/3.4287	Sua (N)	A(1), B(2), C(1), D(4)	8
	Lastourville & Mekambo/G	12.952/−0.5345 & 13.9396/0.9618	Sua (S)	C(1), D(2)	3

(*) C = Cameroon, RC = Republic of the Congo, G = Gabon; GPS coordinates correspond to locations of original populations. (**) Genetic groups based on microsatellite markers refer to the assignment of the different populations inside a larger genetic groups [10]. Ivo = *ivorense*; SuaN = *suaveolens* central North, SuaS = *suaveolens* central South based on reference [16].

**Table 2 molecules-26-01668-t002:** Molecular assignment details of the 28 features identified in cassane-diterpene cluster by molecular network (see Figure 7). The color in the last column are: Sua * in pink, Ouesso-Sua in violet, ivo-Libr1 in light blue, ivo-Libr2 in dark blue as in Figure 5 and Figure 6.

Scaffold					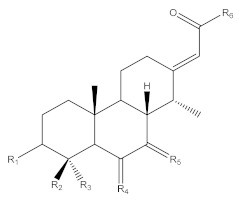						
Number	Molecular Formula/RT (min)	*m*/*z*/δ (ppm)	R_1_	R_2_	R_3_	R_4_	R_5_	R_6_	Ref	More Abundant in:	Relative Abundance in Ouesso/ Sua */ ivo-Libr 1/ ivo-Libr 2
[A1]	C_24_H_37_NO_5_/16.97	420.2733/2.74	H	COOCH_3_	CH_3_	H,H	O	OCH_2_CH_2_ NHCH_3_	[17,32,33,34]	Sua *	9/48/33/11
[A2]	C_23_H_35_NO_5_/16.75	406.2569/4.69	H	COOCH_3_	CH_3_	H,H	O	NHCH_2_CH_2_OH/OCH_2_CH_2_NH_2_	NEW	Sua *	21/71/4/4
[A3]	C_24_H_35_NO_5_/14.54	434.2549/2.74	H	COOCH_3_	CH_3_	O	O	OCH_2_CH_2_NHCH_3_	[32,34]	Ouesso-Sua	53/36/6/5
[A4]	C_25_H_39_NO_5_/17.16	434.2855/10.59	H	COOCH_3_	CH_3_	H,H	O	OCH_2_CH_2_N(CH_3_)_2_	[31,37,41,52]	Sua *	24/52/14/11
[A5]	C_24_H_35_NO_6_/15.77	434.2555/4.11	H	COOCH_3_	CH_3_	O	O	OCH_2_CH_2_NHCH_3_	[32,34]	Ivo-Libr 2	9/11/28/53
[B1]	C_26_H_39_NO_8_/14.44	494.2757/1.74	OCOCH_3_	COOCH_3_	CH_3_	OH (or O)	O (or OH)	OCH_2_CH_2_NHCH_3_	[34]	Sua *	10/82/3/5
[B2]	C_24_H_37_NO_7_/12.62	452.2664/4.7	OH	COOCH_3_	CH_3_	OH (or O)	O (or OH)	OCH_2_CH_2_NHCH_3_	[43,46]	Ouesso-Sua	45/26/19/9
[B3]	C_25_H_37_NO_8_/14.34	480.2607/7.32	OCOCH_3_	COOCH_3_	CH_3_	OH (or O)	O (or OH)	OCH_2_CH_2_NH_2_	NEW	Sua *	16/83/0/0
[B4]	C_25_H_37_NO_8_/14.51	480.2828/49	OCOCH_3_	COOCH_3_	CH_3_	OH (or O)	O (or OH)	OCH_2_CH_2_NH_2_	NEW	Sua *	24/70/4/3
[B5]	C_27_H_41_NO_8_/14.40	508.2980/10.64	OCOCH_3_	COOCH_3_	CH_3_	OMe (or O)	O (or OMe)	OCH_2_CH_2_NHCH_3_	[43]	Ivo-Libr 2	2/4/39/55
[B6]	C_30_H_45_NO_8_/17.04	548.3212/1.09	OCOC(CH_3_)=CHCH_3_ (tigloyl) or OCOCH=C(CH_3_)_2_ (methylcrotonyl)	COOCH_3_	CH_3_	OMe (or O)	O (or OMe)	OCH_2_CH_2_NHCH_3_	NEW	Ivo	0/0/48/52
[B7]	C_30_H_47_NO_8_/17.28	550.3350/ 4.45	OCOCH(CH_3_)CH_2_CH_3_ or OCOCH_2_CH(CH_3_)_2_	COOCH_3_	CH_3_	OMe (or O)	O (or OMe)	OCH_2_CH_2_NHCH_3_	NEW	Ivo	0/0/50/50
[B8]	C_30_H_47_NO_9_/14.05	566.3312/ 2.69	OCOC(OH)(CH_3_)CH_2_CH_3_ or OCOCH(OH) CH(CH_3_)_2_	COOCH_3_	CH_3_	OMe (or O)	O (or OMe)	OCH_2_CH_2_NHCH_3_	NEW	Ivo	0/1/51/48
[B9]	C_28_H_43_NO_7_/15.65	506.3102/ 2.04	OCOC(OH)(CH_3_)CH_2_CH_3_ or OCOCH(OH) CH(CH_3_)_2_	CH_3_	CH_3_	O	O	OCH_2_CH_2_NHCH_3_	NEW	Ivo	0/0/52/48
[B10]	C_28_H_41_NO_6_/17.87	488.2988/ 3.83	OCOC(CH_3_)=CHCH_3_ or OCOCH=C(CH_3_)_2_	CH_3_	CH_3_	O	O	OCH_2_CH_2_NHCH_3_	NEW	Ivo-Libr 1	1/1/59/39
[B11]	C_30_H_48_NO_9_/14.17	566.3308/2.76	OCOC(OH)(CH_3_)CH_2_CH_3_ or OCOCH(OH) CH(CH_3_)_2_	COOCH_3_	CH_3_	OMe (or O)	O (or OMe)	OCH_2_CH_2_NHCH_3_	NEW	Ivo-Libr 2	1/1/3/95
[B12]	C_31_H_49_NO_12_/10.40	628.3323/0.72	O-glucopyrannosyl	COOCH_3_	CH_3_	OMe (or O)	O (or OMe)	OCH_2_CH_2_NHCH_3_	NEW	Ivo-Libr 2	1/0/35/65
[B13]	C_23_H_37_NO_4_/13.27	392.2802/1.70	OH	CH_3_	CH_3_	H,H	O	OCH_2_CH_2_NHCH_3_	NEW	Ivo-Libr 1	4/7/63/27
[B14]	C_23_H_37_NO_4_/12.66	392.2789/1.62	OH	CH_3_	CH_3_	O	H,H	OCH_2_CH_2_NHCH_3_	NEW	Ivo-Libr1	5/6/62/27
[B15]	C_28_H_45_NO_6_/15.83	492.3314/1.15	OCOC(OH)(CH_3_)CH_2_CH_3_ or OCOCH(OH) CH(CH_3_)_2_OCOCH_2_C(OH) (CH_3_)_2_	CH_3_	CH_3_	H,H	O	OCH_2_CH_2_NHCH_3_	[26]	Ivo-Libr1	1/3/53/43
[B16]	C_27_H_43_NO_6_/15.61	478.3067/20.1	OCOC(OH)(CH_3_)CH_2_CH_3_ or OCOCH(OH) CH(CH_3_)_2_OCOCH_2_C(OH) (CH_3_)_2_	CH_3_	CH_3_	H,H	O	OCH_2_CH_2_NH_2_	NEW	Sua *	21/61/7/11
[B17]	C_22_H_35_NO_4_/13.10	378.2625/3.67	OH	CH_3_	CH_3_	H,H	O (or OH)	OCH_2_CH_2_NH_2_	NEW	Sua *	33/62/3/2
[C1]	C_24_H_37_NO_7_/11.17	452.2636/1.50	OH	COOCH_3_	CH_3_	OH (or O)	O (or OH)	OCH_2_CH_2_NHCH_3_	[43,46]	Ouesso-Sua	60/36/1/3
[C2]	C_23_H_35_NO_7_/10.97	438.2501/3.36	OH	COOCH_3_	CH_3_	OH (or O)	O (or OH)	OCH_2_CH_2_NH_2_	NEW	Ouesso-Sua	60/40/0/0
[C3]	C_25_H_39_NO_7_/11.31	466.2787/2.64	OH	COOCH_3_	CH_3_	OH (or O)	O (or OH)	OCH_2_CH_2_NH(CH_3_)_2_	NEW	Ouesso-Sua	69/24/2/5
[BS1]	C_24_H_35_NO_7_/12.84	450.2477/1.27 (Biosigner 450.2492/2.07)	OH	COOCH_3_	CH_3_	O	O	OCH_2_CH_2_NHCH_3_	NEW	Sua *	40/59/0/1
[BS2]	C_28_H_43_NO_6_/18.48	490.3145/3.71 (Biosigner 490.3158/1.05)	OCOC(CH_3_)=CHCH_3_ or OCOCH=C(CH_3_)_2_	CH_3_	CH_3_	OH (or O)	O (or OH)	OCH_2_CH_2_NHCH_3_	NEW	Ivo-Libr 1	0/0/65/35
[BS3]	C_28_H_43_NO_7_/15.65	506.3102/2.04 (Biosigner 506.3113/0.14)	OCOC(OH)(CH_3_)CH_2_CH_3_ or OCOCH(OH) CH(CH_3_)_2_	CH_3_	CH_3_	O	O	OCH_2_CH_2_NHCH_3_	NEW	Ivo	0/0/52/48

## Data Availability

Data are available at this link when you are logged on W4M platform: https://workflow4metabolomics.usegalaxy.fr/histories/list_published under name “Erythro Clean 01” by author F. Souard.

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
