# Peer review of "Does the Phytochemical Diversity of Wild Plants Like the *Erythrophleum genus* Correlate with Geographical Origin?"

_molecules, 2021, doi:10.3390/molecules26061668_

Round 1
Reviewer 1 Report
The manuscript by Delprte et al tries to correlate metabolites of two plant species with their geographical origin. The manuscript is in line with the scope of the journal and very well structured. The command of the English language is above par and novelty is quite high. Still, I have some concerns about the formatting of the manuscript and the design of the experiment. In terms of format, I would suggest the authors delve further in detailing the results in the abstract. In its current form it does not really inform much in terms of the found new compounds or the results that led to the determination of the correlation between the origin and the metabolites.
In terms of the design of experiments, I would ask the authors for a clarification on the following: Plant metabolites are defence mechanisms for plants while having other functions tied to the secondary metabolism of plants. They are influenced, by numerous variables, namely climate (including precipitation, sun exposure, wind, moisture in the air.... and many other variables). These variables are quite different in each year, although following a certain trend. Thus, reading through the manuscript I stayed with the impression that samples were only collected in one year, thus showing a fingerprint of the molecules that were produced due to all the different variables of that year. Even in the experiment that used seeds and waited for their maturity for analysis, I can't help but wonder if those seeds did show any of this variability from the specific conditions of the year they were developed in the tree. Thus, the bottom line is that I am not comfortable with the conclusions sought in this manuscript due to only having samples of one year. If I'm wrong please bring this to my attention, but I really believe this is a flaw of the study and should (could?) it be overcome by testing at least 3 to 5 years of the samples. I am asking for a major review by the authors for them to either present this data or to satisfactorily clarify this conundrum to then post my final recommendation.
Reviewer 2 Report
Specific comments
Some affiliation are written/ numbered several times – please revise and re-number them accordingly to the instruction for authors
L145 – please include a reference number to “Ernst et al.” mentioned here. Moreover it seems that the indicated reference is not in the references list – please revise
L153 – 185 – please remove from Results section, while this information is much more suitable to Materials and methods section (some of information already provided in the M&M section – please revise). In my opinion Table 1 and Figure 2 are more suitable to Materials and methods section; moreover the caption of Figure 2 is too long and is repeating some of the information already provided in the M&M section – please revise
L525-526 – please revise the total number of samples (?! E. ivorense 23 samples, E. suaveolens 21 samples = 44 samples …) while “a homogeneous pool of 41 mature leaf samples from 2 species of Erythrophleum 525 (E. ivorense 23 samples, E. suaveolens 21 samples)”
L641 – while it is well know that sun / temperature may alter their composition , please mention the duration and temperature of the air drying procedure; was it sun exposed?
L729 -731 – please revise the paragraph arrangement
Round 2
Reviewer 1 Report
The manuscript is suitable for publication. Have been clarified.